# Chemical-Physical Characterization of a Binary Mixture of a Twist Bend Nematic Liquid Crystal with a Smectogen

**Abir Aouini [1,2]**, **Maurizio Nobili [2]**, **Edouard Chauveau [2]**, **Philippe Dieudonné-George [2]**, **Gauthier Damême [3]**, **Daniel Stoenescu [1]**, **Ivan Dozov [4]** and **Christophe Blanc [2,\*]**

[1]   IMT-Atlantique, 29238 Brest, France; aouini.abir@hotmail.com (A.A.);
      daniel.stoenescu@imt-atlantique.fr (D.S.)
[2]   Laboratoire Charles Coulomb, Université de Montpellier, CNRS, 34095 Montpellier, France;
      maurizio.nobili@umontpellier.fr (M.N.); edouard.chauveau@umontpellier.fr (E.C.);
      philippe.dieudonne-george@umontpellier.fr (P.D.-G.)
[3]   Physique des Systèmes Complexes, Université de Picardie Jules Verne, 80039 Amiens, France;
      gauthierdameme@gmail.com
[4]   Laboratoire de Physique des Solides, Université Paris-Sud, 91405 Orsay, France; ivan.dozov@u-psud.fr
\*    Correspondence: christophe.blanc@umontpellier.fr

**Abstract:** Nematic twist-bend phases ($N_{TB}$) are new types of nematic liquid crystalline phases with attractive properties for future electro-optic applications. However, most of these states are monotropic or are stable only in a narrow high temperature range. They are often destabilized under moderate cooling, and only a few single compounds have shown to give room temperature $N_{TB}$ phases. Mixtures of twist-bend nematic liquid crystals with simple nematogens have shown to strongly lower the nematic to $N_{TB}$ phase transition temperature. Here, we examined the behaviour of new types of mixtures with the dimeric liquid crystal [4',4'-(heptane-1,7-diyl)bis(([1',1''-biphenyl]4''-carbo-nitrile))] (CB7CB). This now well-known twist-bend nematic liquid crystal presents a nematic twist-bend phase below T ≈ 104 °C. Mixtures with other monomeric alkyl or alkoxy -biphenylcarbonitriles liquid crystals that display a smectic A (SmA) phase also strongly reduce this temperature. The most interesting smectogen is 4'-Octyl-4-biphenylcarbonitrile (8CB), for which a long-term metastable $N_{TB}$ phase is found at room and lower temperatures. This paper presents the complete phase diagram of the corresponding binary system and a detailed investigation of its thermal, optical, dielectric, and elastic properties.

**Keywords:** liquid crystals; nematic twist-bend; binary mixtures; nematic elastic constants

## 1. Introduction

The nematic phase, N, where mesogenic molecules possess long-range orientational order along a direction (called the director **n**) is one of the simplest liquid crystalline states and is also the most used one for electro-optic applications. Several types of nematic states other than the basic N have been identified during the more than a century-long history of liquid crystals (LCs). Their local structure cannot be distinguished from the N phase, but they present some additional features at a larger scale, such as the continuous twist of the director along the helix of the cholesteric (N\*) phase, for example. One of the last identified nematic phase [1–4] is the twist-bend nematic ($N_{TB}$) phase, whose structure was proposed many years before its discovery [5,6]. It is currently mostly accepted that the mesogens of an $N_{TB}$ phase also spontaneously self-assemble into a helical orientational ordering with a rather short pitch length (a few nanometres). The simultaneous twist and bend deformation, however, defines an

intermediate tilt angle $0 < \theta < \pi/2$ between the director and the direction of the helix axis [4,6,7]. With achiral molecules, the handedness of the local structure is doubly degenerate, and domains of opposite chirality are formed.

If well aligned, these domains exhibit electro-optic response times down to the microseconds [8,9] under the application of a DC field perpendicular to the helix. The potential use of this phenomenon in actual electro-optic devices is, however, limited due the thermal properties of the compounds. The occurrence of an $N_{TB}$ phase has been evidenced now in many compounds, mainly with dimeric, trimeric, tetrameric, or bent-core molecules [10]. Apart from a few exceptions [11,12], the N-to-$N_{TB}$ phase transition in pure compounds is found much above room-temperature. Even more problematically, the twist-bend nematic has often a monotropic behaviour or, at best, displays a very limited temperature range of stability [13]. Supercooling is present in most compounds and permits the studies at lower temperatures, but crystallization tends to occur within a few hours. Long-lived metastable $N_{TB}$ phases at room temperature are thus quite rare, even if recent synthesized compounds have shown to exhibit a wider $N_{TB}$ temperature range [14] or even to form a glassy $N_{TB}$ phase rather than to crystallize [15].

Using mixtures is another way to lower the $N_{TB}$ temperature range. It was first used shortly after the discovery of an $N_{TB}$ phase for the dimer 4′,4‴-(nonane-1,9-diyl)bis(([1,1′-biphenyl]-4-carbonitrile)) (CB9CB) [16]. When mixed with the widely-used 4′-pentyl-4-biphenylcarbonitrile (5CB), a simple monomeric nematogen, a strong lowering of the N-to-$N_{TB}$ phase transition is observed from 104 °C for pure CB9CB to 50 °C for 40 wt % of 5CB. Other similar binary mixtures were studied later [17,18] in detail and showed that the $N_{TB}$ phase was usually quite robust against the addition of linear mesogens of the same chemical family or could even be stabilized by them. In one case at least, an $N_{TB}$ phase was also induced in a binary mixture [19]. Problems related to metastability for long term applications, however, remain present with mixtures. They are even amplified since a crystallization in a binary mixture might yield demixing. Note, however, that a very recent (and specific) mixture obtained with a photoreactive monomer has been shown to promote the stabilization of a metastable $N_{TB}$ phase at room temperature by forming a solid network after photopolymerization [20].

Among the compounds that have been mixed with twist-bend nematic liquid crystals, nematogens have been mostly employed. The fact that smectogens have been rarely studied [17,21] might be due to a relative lack of data about the interplay between the $N_{TB}$ phase and smectic phases. The $N_{TB}$ phase shares many textural features, such as focal conic domains, with the smectic-A (SmA) phase due to its pseudo-layer structure [22]. It has even been mistaken for this latter phase in several liquid crystals [2,23] for many years. A very limited number of pure compounds, however, shows the presence of both smectic and $N_{TB}$ phases [10]. Until recently, only a single fluorinated compound was reported to exhibit the three mesophases: nematic, twist-bend, and SmA [24,25]. Since then, syntheses of various homologue series [26,27] have revealed that the $N_{TB}$ phase tends to be replaced by smectic phases in the phase sequence below the nematic phase. The state can be either a SmA phase or a more complex heliconical smectic phase, also predicted by Dozov [6], in which a short-pitch helical structure is present. Mixing a smectogen and a twist-bend nematic liquid crystal could then be a simple way to enforce the coexistence of SmA and $N_{TB}$ phases in a binary system and induce the elusive phase transition between them.

In this paper, we examined the behaviour of [4′,4′-(heptane-1,7-diyl)bis(([1′,1″-biphenyl]4″-carbonitrile))] (CB7CB) when it is mixed with smectogenic monomers. CB7CB is the first dimeric compound where the nematic twist-bend phase was unambiguously identified [2]. It has been studied in detail and its properties thoroughly investigated [4]. We focused on its mixing with other well-known monomeric alkyl or alkoxy–biphenylcarbonitriles and especially with 4′-n-octyl-biphenyl-4-carbonitrile (8CB). One objective was to specify the influence of a smectic region in the binary phase diagrams and to compare with the case of the nematogen 4′-n-pentyl-biphenyl-4-carbonitrile (5CB), whose mixing with CB7CB has been studied in detail [4,17,18,28,29]. Aside from the complete phase diagram of 8CB/CB7CB binary system, we investigated its thermal, optical, dielectric, and elastic properties. Another interest of mixtures is indeed the possibility to continuously modify their physical properties.

The second objective was then related to the evolution of the properties of the nematic phase in such binary systems. In particular, we focused on the question of the bend elastic constant in a system when $N_{TB}$ and SmA phases are present. Although both phases have layered or pseudo-layered structures, the bend constant $K_{33}$ of the nematic phase is expected to behave very differently in the vicinity of the corresponding phase transitions. $K_{33}$ is seen to diverge in smectogens (such as 8CB) when the smectic phase is approached [30,31] (bend deformation is not permitted in a set of parallel layers). On the contrary, the N-to-$N_{TB}$ phase transition has been predicted or explained by the vanishing of $K_{33}$ [6] or of its renormalized version [32,33] in agreement with measurements revealing a strong decreasing of $K_{33}$ when approaching the $N_{TB}$ phase in pure compounds [3,29,34].

## 2. Materials and Methods

### 2.1. Materials

The liquid crystal 1,7-bis(4-cyanobiphenyl-4-yl) heptane (CB7CB) was present in all studied binary mixtures. It is a well-studied LC [2,13,18] that displays an $N_{TB}$ phase below 104 °C. Below T = 102 °C, the $N_{TB}$ phase becomes metastable and tends to crystallize at long times, even in thin cells. We explored the mixtures of CB7CB with different smectogens, mostly 4′-Octyl-4-biphenylcarbonitrile (8CB) but also 4′-Decyl-4-biphenylcarbonitrile (10CB) and 4′-(Octyloxy)-4-biphenylcarbonitrile (8OCB). All three mesogens have similar chemical groups and display a Smectic A phase. The molecules were obtained from Synthon Chemicals GmbH & Co (Bitterfeld-Wolfen, Germany) except for 8OCB (BDH Chemicals, Ltd., Poole, UK).). The chemical formulas and the phase temperature transitions of the compounds are given in Figure 1.

**Figure 1.** Chemical structure and phase transition temperatures (°C) of 4′-Octyl-4-biphenylcarbonitrile (8CB), 4′-Decyl-4-biphenylcarbonitrile (10CB), 4′-(Octyloxy)-4-biphenylcarbonitrile (8OCB) and [4′,4′-(heptane-1,7-diyl)bis(([1′,1″-biphenyl]4″-carbo-nitrile))] (CB7CB) liquid crystals.

### 2.2. Optical Microscopy

Optical observations were performed using either commercial liquid crystal cells (5 μm thickness, polyimide coating from EHC. Co., Ltd., (Tokyo, Japan) or homemade ones with polyvinyl alcohol (PVA) rubbed anchoring layers providing a planar alignment [35,36]. We also used glass substrates silanized with dimethyloctadecyl [3-(trimethoxysilyl) propyl] ammonium chloride (DMOAP), which usually provides a homeotropic alignment for n-alkyl-cyanobiphenyl systems [37]. The cells were mounted by assembling two treated substrates (in an antiparallel configuration for the brushed polymers) using epoxy glue. Their thicknesses were carefully measured using a spectrometer (UV-1205 from Shimadzu

Corporation, Kyoto, Japan) before capillary filling on a Kofler bench. Polarization optical microscopy (POM) observations were made using a LABORLUX 12 POL S microscope (Leitz, Wetzlar, Germany) equipped with a 1024 × 768 digital camera (XCD-X710 from Sony Corporation, Tokyo, Japan), a color camera (D90 from Nikon Corporation, Tokyo, Japan), and a HS400 heating and cooling stage with its STC200D controller (Instec, Boulder, CO, USA, 0.1 °C regulation). Temperature under microscope ranged from −20 °C to 130 °C.

Contact experiments were carried out in homemade 20 × 20 mm$^2$ PVA cells of thickness about 5 μm opened on the four sides. The cells were filled by capillarity at 130 °C with the isotropic phase of CB7CB and the other examined compound (deposited on two opposite sides). An interface formed between the isotropic liquid. Due to the miscibility and the interdiffusion of the molecules, a spatial gradient of concentration was then established. The width of this interface increased with time, but it evolved quite slowly after a few minutes due to the concomitant decrease of the concentration gradient. This method was used to establish rapidly qualitative binary phase diagrams since the concentration of CB7CB ranged from 0 wt % to 100 wt % across this interface. The different states of the phase diagrams could therefore be detected when the temperature was changed.

### 2.3. Birefringence Measurements

Quantitative birefringence measurements were performed at 546 nm with an Abrio System (CRI Inc., Chantilly, VA, USA) installed on a Leica 2500P microscope. This set-up allowed a rapid determination of the optical axis and the retardation for each pixel of the cooled CCD camera. A typical measurement consisted in choosing a region (of typical size 50 × 50 μm$^2$) that remained perfectly planarly aligned in the N$_{TB}$ phase. After heating in the nematic phase close to the isotropic phase, the sample was cooled downed at 0.1 °C. min$^{-1}$ while measuring the mean retardation of the region.

### 2.4. X-Ray Characterization

The different phases were identified by X-ray scattering in a wide-angle configuration (WAXS). A high brightness low power X-ray tube coupled with aspheric multilayer optics (GeniX3D from Xenocs SA, Grenoble, France) was employed. It delivered an ultralow divergent beam (0.5 mrad, λ = 0.15418 nm). Scatterless slits were used to give a clean 0.6 mm beam diameter with a flux of 35 Mphotons/s on the sample. We worked in a transmission configuration, and scattered intensity was measured by a 2D "Pilatus" 490 × 600 pixels detector (Dectris, Baden-Daettwil, Switzerland) with pixel size of 172 × 172 μm$^2$, at a distance of 0.2 m from the sample. Liquid crystal samples were studied in Lindeman glass capillaries (diameter 1.5 mm) filled by capillarity using a Kofler bench and sealed by flame. Scattering intensities were corrected by subtracting the empty cell contribution taking into account the different transmissions.

### 2.5. Differential Scanning Calorimetry

Differential scanning calorimetry (DSC) measurements were performed using a Q2000 DSC system (TA instruments, New Castle, DE, USA)). Heating-cooling cycles were carried out at a constant temperature rate of 10 °C.min$^{-1}$ with a 10 min isotherm between heating and cooling. The samples had a mass between 5 and 7 mg and were sealed in aluminium pans and kept in nitrogen atmosphere during measurement. Thermal data were extracted from the second cooling trace. The enthalpy values $\Delta H$(J/g) were obtained by the integration of the peak of the transition using TA Universal Analysis 2000 software (TA instruments, New Castle, DE, USA, Version 4.5A).

### 2.6. Dielectric and Elastic Characterization

Dielectric measurements and the determination of bend and splay elastic constants of the nematic phase were performed using a homemade system based on a TTi TGA12101 arbitrary wave generator (Thurlby Thandar Instruments Ltd., Huntingdon, UK), a Krohn-Hite 7600 wideband voltage amplifier (Krohn-Hite, Brockton, MA, USA), and a Tektronix DPO3014 oscilloscope (Tektronix Beaverton,

OR, USA), all driven with a PC with a LabVIEW code. Our approach was based on the capacitance technique [30,38] modified as follows. A planar liquid crystal cell (10 μm thick, from EHC Japan) was filled by capillarity in the isotropic phase and connected in series with a load resistance $R_c$. The circuit was submitted to an AC voltage with frequency $f$ and variable amplitude $U_g$. The complex impedance of the cell, $Z$, was then obtained by measuring the amplitude and the phase of the voltage drop, $U_c$, on the load resistance. A first series of measurements was performed at fixed small, applied voltage (below Freedericksz threshold) with variable values of $f$ and $R_c$. This first step allowed the detailed characterization of the equivalent $RC$ circuit of the LC cell and provided the internal cell parameters (LC resistivity, resistance of the indium tin-oxide (ITO) electrodes, and capacitance of the alignment layers). These parameters were then used to determine accurately the dependence of the liquid crystal layer capacitance $C$ on the applied voltage $U$ when varying $U_g$. In a second step, the $C(U)$ curves were measured across the whole nematic phase range by cooling from the isotropic state with $f$ and $R_c$ values fixed at their optimal values defined during the first step. The $C(U)$ curves were then fitted to the expected theoretical dependences [30,39–41] with the LC elastic and dielectric constants as fitting parameters but also taking as additional free parameters the pretilt angle and the surface anchoring strength. The detailed description of this technique will be published elsewhere [42].

## 3. Results and Discussions

### 3.1. Phase Diagrams

The 8CB\CB7CB phase diagram (see Figure 2) was obtained from POM observations, complemented by XRD and dielectric measurements. In polarization optical microscopy (POM), cells were cooled down from 120 °C at −1 °C/ min in order to detect the phase transitions.

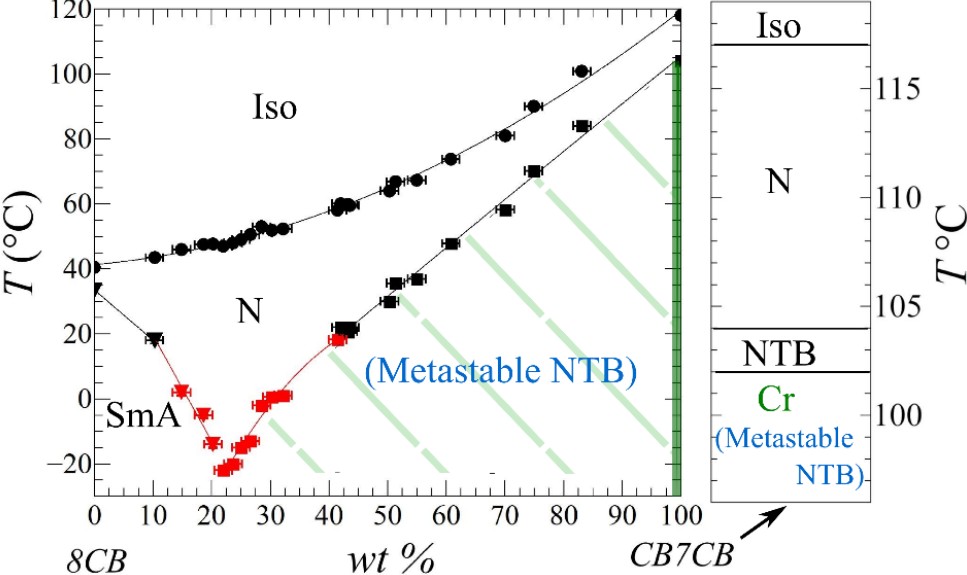

**Figure 2.** Phase diagram of 8CB/CB7CB binary system. The phase transition temperatures were obtained either by (POM) (black symbols) or dielectric measurements (red symbols). The nature of the lower temperature phase was confirmed using XRD. The green line indicates the stable crystalline state of CB7CB. $N_{TB}$ phase is metastable almost everywhere, except possibly a few degrees below the N/$N_{TB}$ phase transition. The thermodynamically stable state of green dashed region is a two phase equilibrium between crystalline CB7CB and either $N_{TB}$ in the high temperature corner or N phase at lower temperatures (see main text). Error bars are mainly due to uncertainty in the concentration of small-quantity samples. They correspond to the data dispersion of the phase transition temperatures.

Let us first focus on the peculiar phase sequence of the pure CB7CB material (shown in the right part of Figure 2) and already well described in the literature [2]. On cooling from the isotropic

(Iso), CB7CB exhibited phase transitions: Iso to N at 117 °C and N to $N_{TB}$ at 104 °C (see Figure S1 in supplementary materials). As said in the introduction, $N_{TB}$ phases of many compounds are monotropic and are therefore entirely metastable. The crystalline form of CB7CB, however, melted slightly below the temperature where $N_{TB}$ phase appeared under cooling [2]. This indicates that $N_{TB}$ phase was stable in a narrow temperature range between ~102 °C and ~104 °C. The phase supercooled extensively, and it was possible to maintain it at room temperature during a few days before crystallization occurred. In an unsealed liquid crystal cell, crystallization could be nevertheless provoked at will with a crystal germ, even at high temperatures (see the growth of crystals in typical $N_{TB}$ textures in Figure S2). Finally, note that it was also evidenced that the (metastable) $N_{TB}$ phase of CB7CB evolved to a (metastable) glassy state below ~4 °C [43]. This state is not indicated in Figure 2.

Similar transitions were observed in the mixtures containing a limited amount of 8CB, as shown in Figure 3 for a sample containing 50 wt % CB7CB. Two-phase regions could be observed at the phase transitions when fixing the temperature. They were nevertheless very narrow (typically a few tenths of degrees) and are not visible in Figure 2. Usually, the nature of LC phases and an accurate determination of the transitions temperatures can be obtained from POM observations alone, but here, the properties of phases and phase transitions required several additional techniques. First, the $N_{TB}$ and the SmA textures looked very similar with the same macroscopic defects (focal conic domains) [2]. XRD was then used to identify the nature of the low-temperature phase. Contrary to nematic and $N_{TB}$ phases, the SmA phase is indeed characterized by a sharp Bragg peak (see Figure S3) due to the presence of a positional order inexistent in the two other phases [4]. Furthermore, for concentrations between 15 and 30 wt % of CB7CB, the N/$N_{TB}$ as well as the N/SmA precise phase transitions temperatures were hardly determined by optical means. While the front interface was easily visible outside this range (see Figures 3 and 4a), the nematic fluctuations slowly vanished while decreasing temperature, and a clear front interface was no longer observed. Figure 4b shows this phenomenon at $\phi$ = 30 wt %. For this concentration, the phase transition could still be determined in a narrow temperature interval, but at lower concentrations, the contrast change occurred over several degrees. A similar behaviour was already reported with 5CB/CB7CB mixtures at a high 5CB concentration [17]. The resulting texture remained planar, devoid of defects, while the typical $N_{TB}$ textures appeared rapidly when $\phi \gtrsim 30$ wt % (see Figure 3c). It was only at much lower temperature, after fast cooling, that the typical textures of $N_{TB}$ in planar cells were observed (see Figure S4). Dielectric measurements, however, still showed a sudden increase of the Freedericksz threshold at the N/$N_{TB}$ (or at the N/SmA) phase transition, which was then used to identify more accurately the lower boundary of the N phase with a typical resolution of ~1 °C. The nature of the SmA phase was still confirmed by a sharp X-Ray present below $\phi \approx 20\%$.

As with 5CB [17,18], the addition of 8CB to CB7CB induced a decrease of both Iso/N and N/$N_{TB}$ phase transitions temperatures. It should be noted that adding CB7CB to 8CB also decreased the N/SmA transition temperature but concomitantly increased the Iso/N transition temperature, which yielded a broadening of the N phase. The largest N temperature range (about 60 °C) was observed at $\phi = \phi_c \approx 20$ wt % in a tongue-like region that separated the SmA and the $N_{TB}$ phases. A clear $N_{TB}$/SmA phase transition was thus not observed but rather an extension of N phase down to very low temperatures (~−20 °C). Although $N_{TB}$ and SmA phases displayed very similar textures, a direct phase transition between the two phases was not observed neither with 8CB nor with other cyanobiphenyl-like smectogens. We tested close compounds such as 8OCB with higher phase temperature transitions and 10CB, a smectogen without N phase. Contact experiments performed in liquid crystals cells between CB7CB and one of these mesogens showed the simultaneous presence of SmA and $N_{TB}$ phases separated by a nematic domain, even at room temperature (see Figures S5 and S6). In each case, the N region expanded and separated the $N_{TB}$ and the SmA regions down to the lowest temperatures for which the mixture crystallizes. This is in contrast with a few studies [24,27,44–46] where an Sm/$N_{TB}$ phase transition was reported in pure compounds. In our case, this behaviour, however, promoted the existence of low temperature N and $N_{TB}$ phases in the mixtures. We examined the behaviour of

the $N_{TB}$ phase in contact with a tiny crystal seed briefly deposited at the periphery of an opened cell (see a detailed example in Figure S7). The seed yielded a partial crystallization in the entire cell and the coexistence of crystals with a nematic phase, indicating that the N phase was stable, whereas the $N_{TB}$ phase was only metastable at low temperature. The equilibrium phase diagram of the 8CB/CB7CB binary system, therefore, shows a large two-phase region in place of the $N_{TB}$ domain (see Figure 2). In sealed cells, however, 8CB/CB7CB mixtures in the range 40 wt % < $\phi$ < 55 wt % exhibited room temperature $N_{TB}$ phases that were observed to be metastable for at least 6 months in XRD capillaries and optical cells. It should be noted that, if other studies have already reported low temperature $N_{TB}$ phases in mixtures [16–18,47], many of them tend to rapidly crystallize at room temperature [20]. This feature allowed us to characterize the 8CB/CB7CB system in detail.

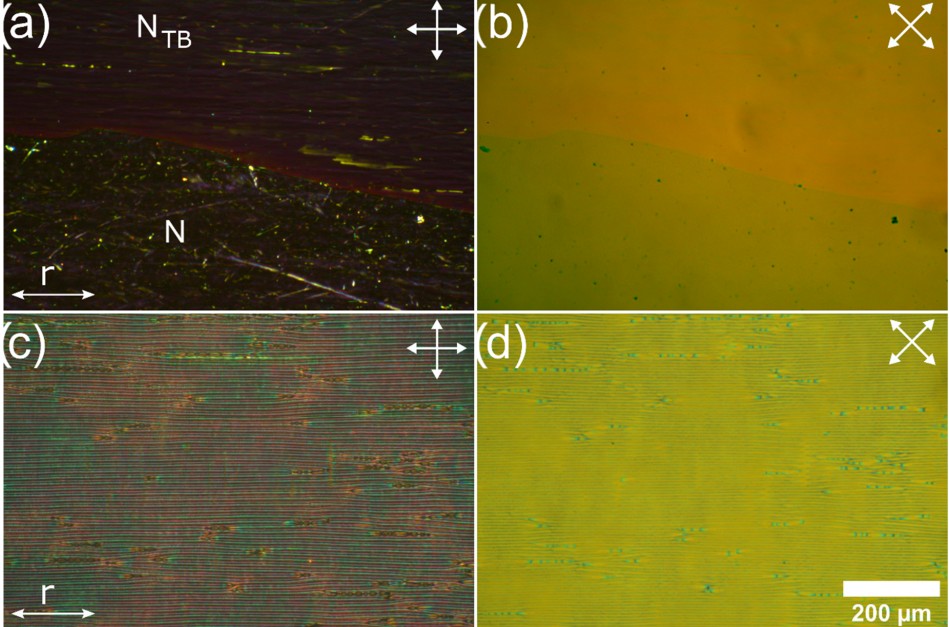

**Figure 3.** Polarized optical micrographs of a CB7CB/8CB mixture (50 wt %/50 wt %) observed (**a**,**b**) at 32 °C, during the N/$N_{TB}$ phase transition and (**c**,**d**) at 31 °C in the $N_{TB}$ phase. Crossed polarizers; r indicates the rubbing direction.

*3.2. Enthalpies of the Phase Transitions*

Adding 8CB to CB7CB was expected to affect the properties of the different phases as well as the various phase transitions. DSC measurements showed a clear heat flow peak at the Iso/N phase transition in the whole range of concentrations $\phi$ (see Figure S8). The N/$N_{TB}$ phase transition peak, on the contrary, rapidly decreased while decreasing $\phi$ and was hardly detectable when approaching $\phi_c$. For pure CB7CB, the enthalpy values of Iso/N (1.76 J/g) and N/ $N_{TB}$ (1.84 J/g) phase transitions were close (see also [18]), but they rapidly differed when adding 8CB. For $\phi$ = 32.5 wt %, the enthalpy value of the Iso/N transition (6.48 J/g) was much higher than the N/$N_{TB}$ one (0.062 J/g), which had strongly decreased. Near $\phi_c$, the latter was quite low (a few tens of mJ/g), indicating an almost second order phase transition (Figure 5). The fact that the Iso/N and the N/$N_{TB}$ enthalpies showed antagonistic evolutions has already been observed and investigated in another mixture containing CB7CB [48]. The second compound was, however, also a twist-bend nematic liquid crystal, and this phenomenon was explained by the different average curvatures of the mesogens in a mean-field Landau approach. It was also observed in a series of pure compounds [45] (difluoroterphenyl-based dimers with alkyl spacers). In that case, it was explained by the role of the spacer length, which facilitates the twist of the molecules and concomitantly reduces the energy difference between N and $N_{TB}$ phases. The authors of reference [45] noted that their molecular approach was compatible with the results of reference [48],

assuming an average flexibility in a binary system. The presence of rod-like 8CB molecules in the mixtures we examined could play here the same role for the average mesogen, rendering it less bent.

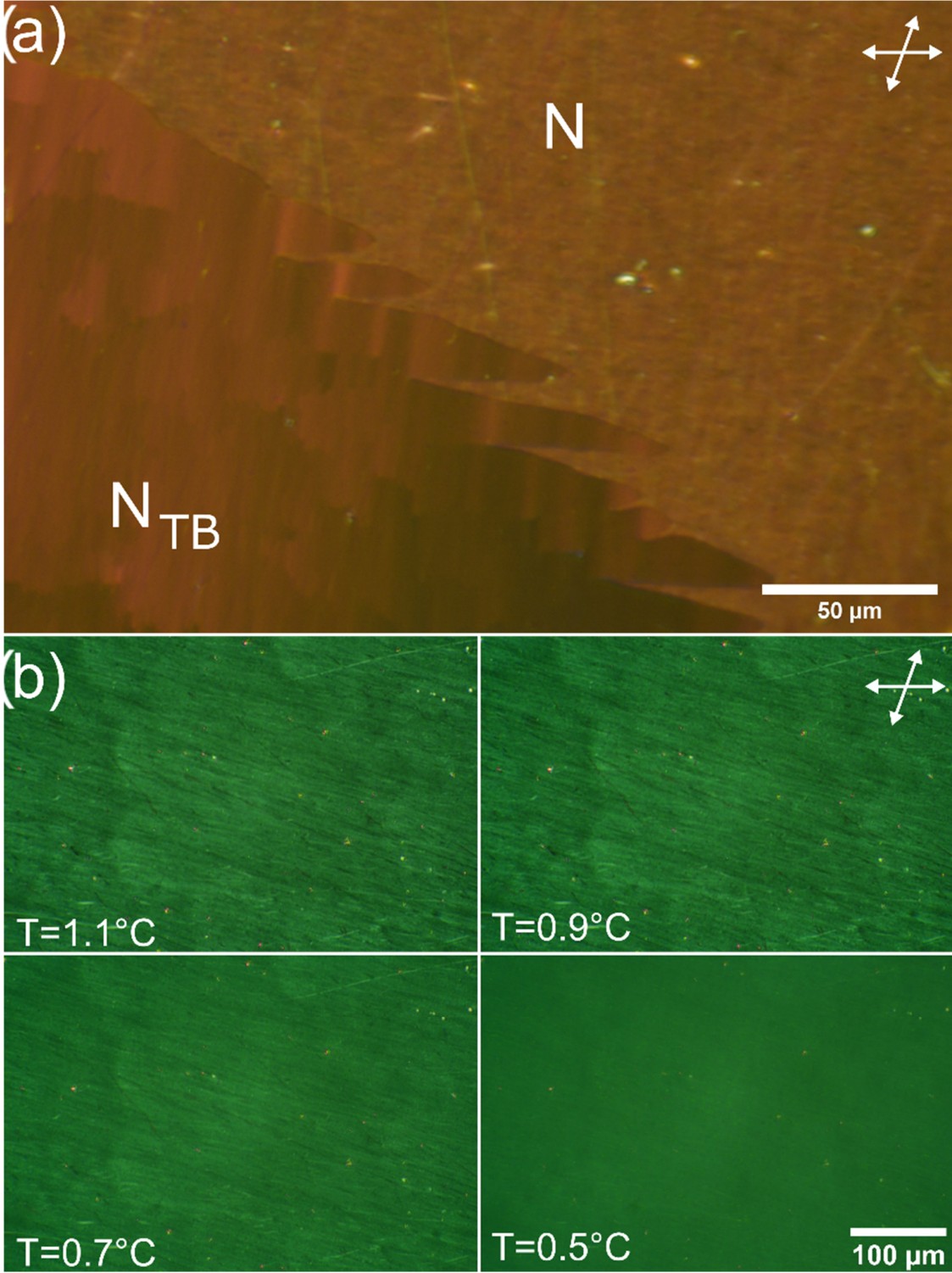

**Figure 4.** Polarized optical micrographs of the 8CB/CB7CB binary system during the N/N$_{TB}$ phase transitions at two different concentrations. (**a**) $\phi = 55$ wt %: a front interface between the two phases is easily observed at T = 37 °C. (**b**) $\phi = 30$ wt %: the phase transition is characterized by a gradual change of contrast (over ~0.5 °C) but without any front interface. Slightly uncrossed polarizers.

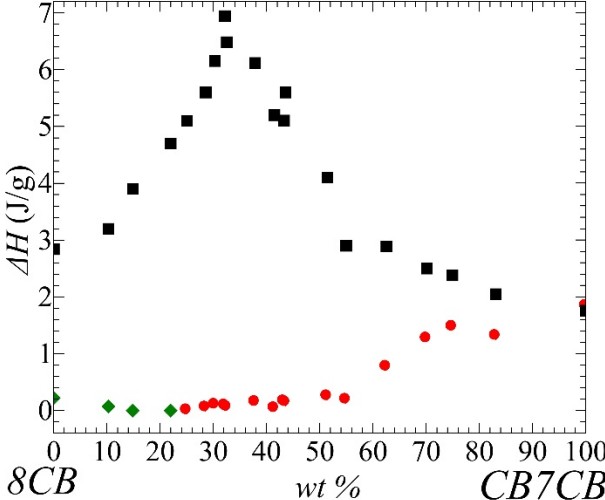

**Figure 5.** Enthalpy of phase transitions as a function of CB7CB concentration. Black squares correspond to the Iso/N phase transition, red disks to the N/ $N_{TB}$ phase transition, and green diamonds to the N/SmA phase transition.

### 3.3. Evolution of Birefringence

The softening of the N/$N_{TB}$ phase transition was also revealed by optical measurements. Birefringence of pure CB7CB as a function of temperature is given in Figure 6a. A similar measurement has already been discussed in [7] and can be summarized as follows. The birefringence $\Delta n$ of the N phase was well-fitted with the classical Haller formula [49]. A slight departure was, however, present a few degrees above the N/ $N_{TB}$ transition, marked by a small jump of the birefringence value [7,17] at the phase transition and characteristic of a first-order phase transition. Under cooling, after the transition, $\Delta n$ decreased rapidly presumably due to the increase of the conical tilt angle $\theta$ of the $N_{TB}$ phase [7]. This behaviour was not found in mixtures with a sizable amount of 8CB. For example, the mixture with $\phi = 43.5$ wt % (Figure 6b) showed a quite different behaviour. The birefringence of the N phase could still be partially fitted by Haller formula at high temperature. However, under cooling, a maximum value was observed at $T \approx 38$ °C, preceding a slight decrease of 0.012 before remaining almost constant. The N/$N_{TB}$ transition was found at much lower temperature ($T \approx 22$ °C) but could be hardly detected by a change of $\Delta n$ (see inset of Figure 6b). The continuity of the birefringence value also tends to indicate a very weak first order transition when approaching $\phi_c$. Birefringence measurements with higher resolution techniques could be useful to analyse in depth the behaviour of $d\Delta n/dT$ and to characterize more quantitatively the evolution of the phase transition order when approaching $\phi_c$.

### 3.4. Anchoring Transition

The evolutions of the $N_{TB}$ textures and the anchoring properties were also very sensitive to the amount of 8CB in the mixtures. Two main phenomena were observed. First, as said above, the typical stripes and rope textures of $N_{TB}$ phases [1,50,51] (Figure S9) were not observed when approaching $\phi_c$. This behaviour could be correlated to the evolution of the birefringence with $\phi$. According to the simple model developed in [7], the $N_{TB}$ birefringence was mainly related to the tilt angle of the local director **n** with respect to the twist-bend helix. A constant birefringence in the vicinity of the phase transition thus tended to indicate that the tilt angle remained quite small when entering into the $N_{TB}$ phase. This could explain why one can go further into the $N_{TB}$ phase without provoking stripes. These typical textures of the $N_{TB}$ phases are indeed considered to be the consequence of a mechanical Helfrich-Hurault instability [51,52] due to the increase of the heliconal angle and the correlated decrease of the pseudo-layers thickness. Second, the pure CB7CB compound showed a degenerate planar anchoring on DMOAP-treated substrates (Figure 7), as evidenced by the presence of ±1/2 disclinations. Below 50 wt % of CB7CB, the mixtures, however, displayed a perfect homeotropic texture under crossed

polarisers for both N and N$_{TB}$ phases, as shown in Figure 7. It should be noted that homeotropic textures of an N$_{TB}$ phase are quite difficult to achieved in cells. In a few rare cases, they were obtained using somewhat complex methods such as an inorganic passivation layer [53] or with a shear [54,55]. Here, we obtained them simply by applying a classic silane treatment. This evolution is the direct consequence of the presence of 8CB. The self-assembled monolayers of DMOAP indeed give rise to a compact alkyl monolayer on the substrates that favours the perpendicular alignment of the 8CB molecules [56], which possess a pendant alkyl chain contrary to CB7CB molecules. A larger amount of 8CB then drives an anchoring transition from planar to homeotropic.

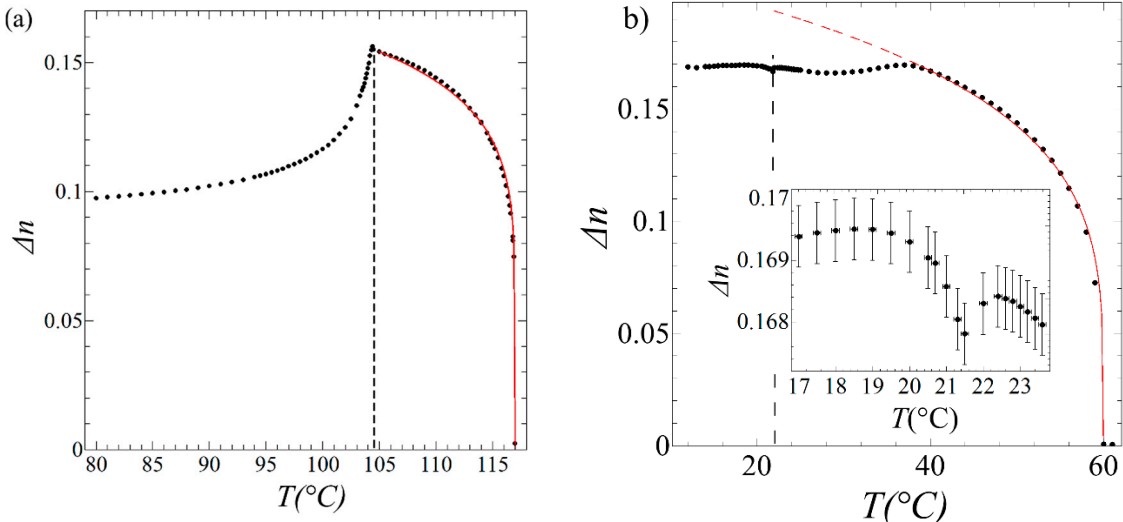

**Figure 6.** Variation of the birefringence with the temperature for (**a**) CB7CB ($\phi$ = 100 wt %), (**b**) a mixture with $\phi$ = 43.5 wt %. In both figures, the plain red curve (extended by a dashed curve) is a fit of the birefringence in the nematic phase using Haller formula [49]. The birefringence of the mixture strongly departs from Haller formula much above the N/N$_{TB}$ phase transition, indicating strong pretransitional effects. The black dashed lines indicate the position of the N/N$_{TB}$ phase transition measured by optical microscopy. A small effect is also seen at the transition in the mixture but is of the order of our resolution. Error bars ($\delta T \sim$ 0.1 °C and $\delta \Delta n \sim$ 0.001) are smaller than symbols in the main panels.

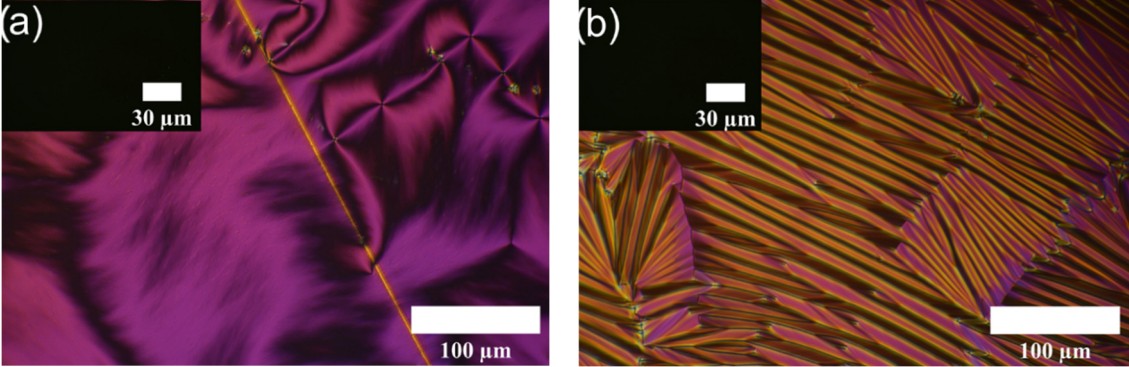

**Figure 7.** In symmetric cells treated with dimethyloctadecyl [3-(trimethoxysilyl) propyl] ammonium chloride (DMOAP), CB7CB displays a degenerate planar anchoring in the N phase (**a**) giving rise to unoriented stripes in N$_{TB}$ phase (**b**). On the contrary, the mixture $\phi$ = 38.42 wt % shows a very good homeotropic alignment in both N ((**a**), **inset**) and N$_{TB}$ ((**b**), **inset**) phases. Polarized optical micrographs, crossed polarizers.

### 3.5. Dielectric Permittivities

The addition of 8CB also strongly modified the dielectric anisotropy and the elastic constants of CB7CB. We examined them as a function of the reduced temperature $T_r$. The latter is defined as the ratio of the temperature $T$ to the Iso/N phase transition temperature $T_{Iso/N}$ (both in Kelvin). Although the Maier–Saupe [57] universal variation of the nematic order parameter $S$ as a function of the reduced temperature is not fully respected in practice, the use of $T_r$ permits to compare values of compounds physicals properties at roughly similar values of $S$. This is interesting especially for the studied system which shows large variations of temperature transitions and ranges of nematic phase existence.

Figure 8 shows the dielectric permittivities of different mixtures in the N phase. The values of the pure compounds are comparable to the values reported in the literature (for 8CB, see [30,49,58] and CB7CB, see [29]). The pure CB7CB has a parallel permittivity ($\varepsilon_\parallel$) value lower than the 8CB but a higher perpendicular one ($\varepsilon_\perp$) [29,30], yielding a rather small dielectric anisotropy. Upon the addition of 8CB, we observe both an increase of $\varepsilon_\parallel$ and a decrease of $\varepsilon_\perp$ concomitantly to the increase of the nematic temperature range. These factors all contribute to a significant enhancement of the dielectric anisotropy $\Delta\varepsilon$ in the N phase, up to $\Delta\varepsilon \approx 10$ when approaching $\phi_c$. For instance, the value of the anisotropy for the mixture of 42 wt % is around 7.2 at room temperature, while that of CB7CB is only 2.2.

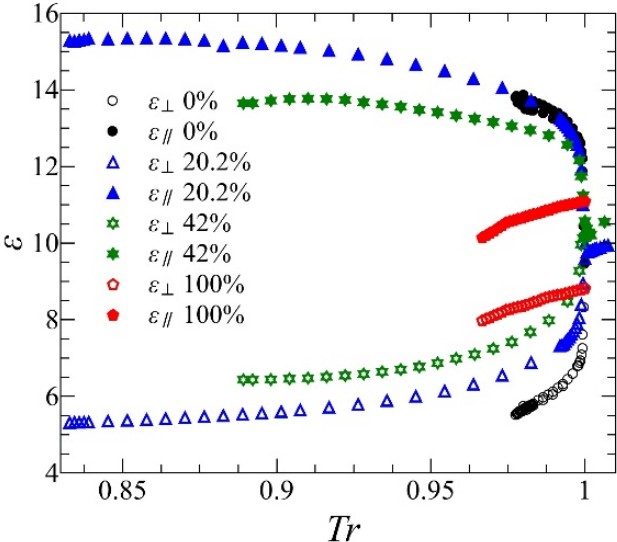

**Figure 8.** Dielectric permittivities of the mixtures as a function of the reduced temperature $T_r$ and for different fraction $\phi$ of CB7CB. Closed symbols represent the parallel dielectric permittivity $\varepsilon_\parallel$ whereas open symbols indicate the perpendicular dielectric permittivity ($\varepsilon_\perp$). Errors bars are smaller than the size of the symbols.

### 3.6. Nematic Elastic Constants

Concerning the elastic constants, while the splay constant did not change too much in the mixtures (Figure 9a), a strong evolution of the bend one is observed (Figure 9b). The pure compounds have very similar splay constants $K_{11}$ that peak at $\approx 8$ pN at their respective lower nematic range. Larger values are observed in some mixtures, but this was mainly due to the broadening of the nematic range. At a given reduced temperature, the splay constant indeed is maximal for the pure compounds. The lowest values of $K_{11}$ are then found for mixtures about 30 wt % of CB7CB and are slightly larger than half the values of the pure compounds. The behaviour of the $K_{33}$ bend constant is very different. The N/SmA transition is, as expected [30], characterized by the divergence of $K_{33}$ (see Figure 9b). On the contrary, cooling decreases the bend modulus of CB7CB to a very low value, about 0.6 pN, about a degree above the N/N$_{TB}$ phase transition before showing a slight increase. This behaviour is now well documented for twist-bend liquid crystals [29,34,59], including CB7CB [60] (reference [29] gives similar but slightly different values with a different technique). More interestingly, in the binary system

studied here, we observe that the values of $K_{33}$ close to the N/N$_{TB}$ phase transition first decrease when 8CB is added (for $\phi = 42$ wt %, we measured $K_{33} \approx 0.3$ pN) before increasing at higher fractions. This surprising behaviour can be related to the very recent observation [61] that the lowest bend constant of a 5CB/CB7CB/CB11CB ternary system is much lower than in CB7CB or CB11CB alone. In our case, however, a careful examination of Figure 9b shows that, at a given value of reduced temperature (~given nematic order parameter), the addition of 8CB continuously increases the value of $K_{33}$, thus the lower values in mixtures could be mainly due to the broadening of the nematic thermal range. This could be also the main explanation of the effect noted in reference [61].

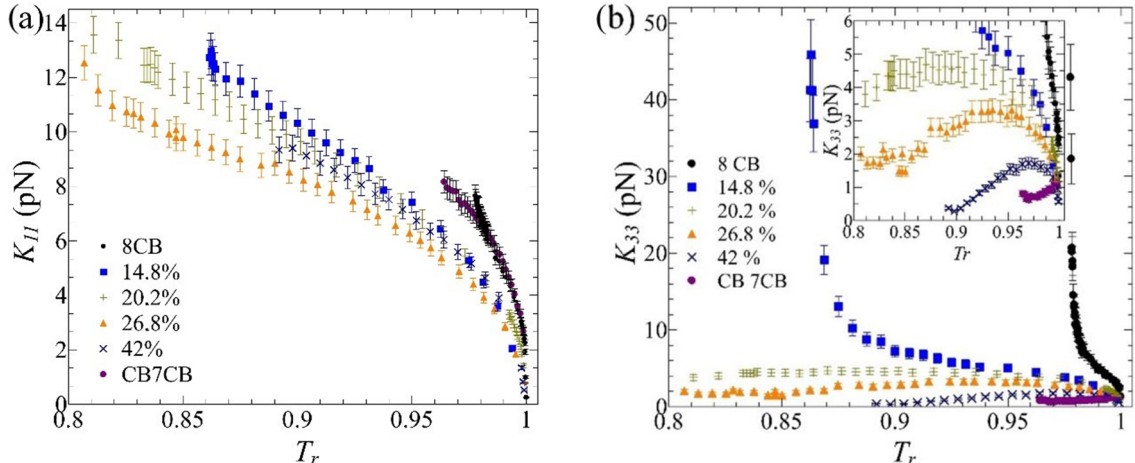

**Figure 9.** Evolution of the (**a**) splay $K_{11}$ and (**b**) bend $K_{33}$ elastic constants of the mixtures as a function of the reduced temperature Tr. **Inset** of (**b**) shows a zoomed region of the $K_{33}$ axis between 0 and 6 pN.

Finally, we note that the change of behaviour of $K_{33}$ from a strong decrease to a divergence is rather progressive. Around $\phi_c$, the bend constant does not decrease towards sub-pN values, even in the very close vicinity of the N$_{TB}$ phase. At $\phi_c$, one can even observe (Figure 9b, $\phi = 20.2\%$) an almost flat value of 4–5 pN over tens of degrees, a perfect illustration of the antagonistic influences of the close SmA and N$_{TB}$ phases. This evolution certainly deserves more attention to clarify the behaviour or the role of $K_{33}$ with the N/N$_{TB}$ phase transition.

## 4. Conclusions

We extensively studied here a binary system consisting of a well-known dimeric twist-bend nematic liquid crystal (CB7CB) mixed with a smectogen (8CB) of the same chemical family. Similarly to simple nematogens, 8CB added to CB7CB strongly reduced the N-N$_{TB}$ phase temperature transition and improved the thermodynamic metastability of the N$_{TB}$ phase. A broader nematic region was obtained, showing the good compatibility between the molecules. The lower phase was either an N$_{TB}$ or a SmA phase, but we were not able to observe a plain N$_{TB}$ to SmA phase transition. Despite their macroscopic resemblance, SmA and N$_{TB}$ domains seem incompatible and are separated by a nematic region down to the lower temperatures. Similar trends were also found with 8OCB and 10CB.

Apart from the stabilization of the twist-bend nematic phase at room temperature, the addition of 8CB to CB7CB yields strong changes. This can be either surface properties such as an easy homeotropic orientation of the N$_{TB}$ phase or bulk properties. In particular, a broad nematic phase with a large dielectric anisotropy and a very low bend elastic constant is present in the centre of the phase diagram. The evolution of $K_{33}$ with the concentration is especially interesting, since it can be easily and continuously tuned on a wide range of values while the splay constant remains almost the same.

**Supplementary Materials:** The following are available online at http://www.mdpi.com/2073-4352/10/12/1110/s1, Figure S1. Polarizing optical micrographs of CB7CB observed at different temperatures in a planar cell. Figure S2. Growth of needle crystals in the N$_{TB}$ phase of CB7CB at 95 °C in a planar cell Figure S3. X-rays diffracted intensities of 8CB/CB7CB mixtures as a function of wave vector *q*. Figure S4. POM of the N$_{TB}$ phase of a 8CB/CB7CB mixture ($\phi = 25$ wt %) at −20 °C. Figure S5. Contact experiments at room temperature of various smectogens-(a) 8OCB, (b) 10CB, (c) 8CB- with CB7CB. Figure S6. Partial phase diagram of the 8OCB/CB7CB binary system. Figure S7. Destabilization at 26 °C of the N$_{TB}$ phase towards needle crystals and a N phase. Figure S8. Heat flow curves of pure CB7CB and of a $\phi = 43$ wt % mixture. Figure S9. POM texture of N$_{TB}$ phase at room temperature of the 8CB/CB7CB binary system at $\phi = 50\%$.

**Author Contributions:** Conceptualization, C.B. and A.A.; methodology, I.D. and C.B.; investigation and experiments, A.A., C.B., E.C., G.D., and P.D.-G.; resources, M.N.; writing—original draft preparation, A.A.; writing—review and editing, C.B.; supervision, C.B., I.D., D.S., and M.N.; project administration, M.N. All authors have read and agreed to the published version of the manuscript.

**Funding:** This research was funded by the Agence Nationale pour la Recherche ANR (France) through grant BESTNEMATICS, ANR-15-CE24-0012.

**Acknowledgments:** We would like to thank Patrick Davidson and Claire Meyer for discussions.

**Conflicts of Interest:** The authors declare no conflict of interest.

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
