# Peer review of "Chemical-Physical Characterization of a Binary Mixture of a Twist Bend Nematic Liquid Crystal with a Smectogen"

_crystals, doi:10.3390/cryst10121110_

Round 1
Reviewer 1 Report
The paper presents results of an extensive study by optical, electro-optical, x-ray and calorimetric techniques of binary mixtures of liquid crystalline materials exhibiting ordinary nematic (N), twist-bend nematic (NTB) and smectic (SmA) mesophases. The authors demonstrate how adding a relatively large proportion of a smectogen into a twist-bend nematic material lowers the temperature of transition into NTB phase, alters the type of surface alignment by standard coatings, increases the dielectric anisotropy and alters the bend elastic modulus.
Liquid crystals with NTB phase indeed have been attracting significant attention throughout the past decade. Such materials are of large fundamental interest and the nature of their microscopic ordering is still under debate. The practical interest is supported by intriguing electro-optical behavior and, in particular, microsecond switching times. In this context, the paper provides an interesting contribution and contains large amount of valuable experimental data. The paper will be of interest for the readers and can be published after the authors clarify the following issues:
- They position the lowering of N-NTB transition temperature as an advantage, but do not provide even a single practical argument against the pure CB7CB material. According to Figure 1, it exhibits a stable NTB phase everywhere below 100 C. This is accompanied by self-contradicting statements, such as that “the twist-bend nematic has often a monotropic behaviour or, at least, displays a very limited temperature range of stability” (lines 50-51). I hope the authors understand that those are opposite rather than close types of behavior. It must be clearly articulated, what really occurs in the bottom-right part of the phase diagram in Figure 1.
- We find another broken tile of the puzzle in lines 212-213: “However, 8CB/CB7CB mixtures, in the range 40wt.<?<55wt. exhibit room temperature NTB phases, that were observed to be metastable for at least 6 months in XRD capillaries and optical cells.” Does this mean that the NTB phase is always metastable?
- The temperature dependencies of birefringence (sec. 3.3) and dielectric anisotropy (sec. 3.5) are considered separately and with different methodology. However, both are similarly defined by the temperature dependence of nematic order parameter. Is it possible to compare the data for the same mixture and extract S(T)?
- In conclusion they write about mixing CB7CB with various smectogens (line 341). However, only one smectogen 8CB has been studied in the main text. More data on other smectogens is provided in Supplementary Materials, which though is really poorly integrated into the paper and seem excessive.
- It will be a good idea to go through the paper and check the wording: “precise” is not a verb (line 87); “authorized” (line 97); “As for 5CB” (line 195), etc.
Reviewer 2 Report
The manuscript entitled " Chemical-physical characterization of binary mixtures of a twist bend nematic liquid crystal with smectogens" by the authors: Abir Aouini, Maurizio Nobili, Edouard Chauveau, Philippe Dieudonné-George, Gauthier Demême, Stoenescu Daniel, Ivan Dozov, Christophe Blanc, is well written and organized. They have used several common techniques to characterize a mixture of twist bend nematic and smectic phases. The authors have focused on broadening the nematic phase down to around room temperature. The stability of this new mixtures has been studied well through thermal, optical, electrical and elastic properties. I only have a few minor comments:
- In the Results and Discussions sections, lines 289 and 290, the sentence needs to be corrected: It sound incomplete as you are defining Tr.
- It would be great if you could state the accuracy of your measurements either in the caption of the figures or as error bars, especially in figure 2, Figure 4, Figure 6 and Figure 7.
I believe this will help liquid crystal community following the work, and I recommend publication after considering these comments as the characterization of this binary mixtures was performed with care and precision.
Reviewer 3 Report
This experimental paper considers binary systems consisting of a dimeric twist bend nematic liquid crystal (CB7CB) mixed with different smectogens. It was shown the compounds added to CB7CB improve the thermodynamic stability of the NTB phase. The complete phase diagram of the studied binary system is presented. It is based on detailed investigations of thermal, optical, dielectric and elastic properties of systems. I find the paper well written, interesting and introducing additional insight into phase behaviour of twist bend LCs. Therefore, I suggest that the paper is accepted.
Reviewer 4 Report
The manuscript reports about the mixture of dimeric liquid crystal CB7CB, which represents the nematic twist bend phase NTB and smectogen 8CB, which exhibits SmA phase at the room temperature. Authors have shown that the addition of 8CB stabilizes the existence of the NTB phase and they report about thermal, optical, dielectric and elastic properties of these mixture and they are presenting a phase diagram of this binary system. The paper is interesting, well organized and clearly written. However, I think it can not be published in the present form. Some explanations of results are not precise and I think authors should address several points before publication:
- The phase diagram presented in Figure 2 does not show also the crystalline phase or eventually the glassy NTB phase so it is not clear what is the real improvement of the thermodynamic stability of the NTB phase. I suggest to add this lower boundary of the NTB phase.
- The phase transition N -> NTB is not presented and justified. The authors said that this transition can not be detected by optical means for concentrations between 15 and 30%, this transition can also not be detected by XRD, heat flow and birefringence measurements, authors said that the transition is determined by the sudden increase of the Freedericksz threshold, but these data are not presented and it is not clear how accurately the temperature range of the NTB phase is determined.
- POM images of the NTB phase and N- NTB transition with planar orientation are not presented in the main text, there is only one image of the NTB texture in the Supplementary. I think it would be nice to present the sequence of images from N to NTB phase for two different concentrations above the critical concentration where the first order of N-NTB phase transition can be clearly observed and below this concentration where is the N-NTB phase transition probably the second order transition.
- It is not clear how “contact experiments” are performed. Is the cell first partially filled with CB7CB and then the smectogen is added? HAs been filling done at room temperature? How and when the nematic phase is established between the NTB and SmA phases?
- I wonder why there is no observable effect in the birefringence measurements at N-NTB phase transition? I think authors should explain this more clearly.
Round 2
Reviewer 1 Report
The authors have clarified all important points, corrected and appropriatele extended the paper.
Reviewer 4 Report
Authors have addressed all my questions and recommendations and the paper
can be published in present form.